# Field-linear anomalous Hall effect and Berry curvature induced by spin chirality in the kagome antiferromagnet Mn$_3$Sn

Xiaokang Li [1] ✉, Jahyun Koo[2], Zengwei Zhu [1] ✉, Kamran Behnia [3] & Binghai Yan [2] ✉

During the past two decades, it has been established that a non-trivial electron wave-function topology generates an anomalous Hall effect (AHE), which shows itself as a Hall conductivity non-linear in magnetic field. Here, we report on an unprecedented case of field-linear AHE. In Mn$_3$Sn, a kagome magnet, the out-of-plane Hall response, which shows an abrupt jump, was discovered to be a case of AHE. We find now that the in-plane Hall response, which is perfectly linear in magnetic field, is set by the Berry curvature of the wavefunction. The amplitude of the Hall response and its concomitant Nernst signal exceed by far what is expected in the semiclassical picture. We argue that magnetic field induces out-of-plane spin canting and thereafter gives rise to nontrivial spin chirality on the kagome lattice. In band structure, we find that the spin chirality modifies the topology by gapping out Weyl nodal lines unknown before, accounting for the AHE observed. Our work reveals intriguing unification of real-space Berry phase from spin chirality and momentum-space Berry curvature in a kagome material.

Understanding the origin of anomalous Hall effect (AHE), observed as early as 1881 in ferromagnetic solids[1] has been enriched in the present century by considering the role played by the topology of electron wave-function[2,3]. In a uniform magnet, the Berry curvature leads to anomalous velocity[4] as a fictitious magnetic field in the momentum-space, which exhibits monopole-like texture[5] around the Weyl point[6,7], and is extensively regarded the intrinsic source of AHE. In unconventional magnetic structures like skyrmions[8], an electron hopping on non-coplanar spin lattice with spin chirality[9–12] picks up a real-space Berry phase and also lead to a Hall response, which is commonly referred to as the topological Hall effect (THE)[13,14]. Both AHE and THE are characterized by a non-linear Hall resistivity. To the best of our knowledge, there is no report on anomalous Hall response without significant departure from field-linearity, believed to be a necessary ingredient for separating ordinary and unusual components of the Hall response. Here, we present a counter-example to this common belief.

Recent theoretical predictions of a large intrinsic AHE in non-collinear antiferromagnets with a nearly compensated magnetization[15–17] was followed by the experimental discovery of sizeable room-temperature AHE in Mn$_3$X (X = Sn and Ge)[18–20] and its counterparts, such as the anomalous Nernst[21,22], thermal Hall[22–24] and magneto-optical Kerr effect[25,26], as well as topological and planar Hall effects[27–30], which appear in presence of the topologically non-trivial domain walls[31] of this magnet. Since the scalar spin-chirality vanishes in the co-planar spin texture, the AHE can solely be understood by the Berry curvature[15–17] with the co-existence of Weyl points[6,32] in the band structure. Potential applications are identified in a variety of fields such as antiferromagnetic spintronics[33–36] and transverse thermopiles[21,37].

[1]Wuhan National High Magnetic Field Center and School of Physics, Huazhong University of Science and Technology, Wuhan 430074, China. [2]Department of Condensed Matter Physics, Weizmann Institute of Science, 7610001 Rehovot, Israel. [3]Laboratoire de Physique et d'Étude des Matériaux (ESPCI—CNRS—Sorbonne Université), PSL Research University, 75005 Paris, France. ✉e-mail: lixiaokang@hust.edu.cn; zengwei.zhu@hust.edu.cn; binghai.yan@weizmann.ac.il

More recently, intriguing AHE was also extensively studied in emerging kagome materials such as $AV_3Sb_5$ (A = K, Rb, Cs)[38,39] and $RMn_6Sn_6$ (R is a rare earth element)[40–42].

$Mn_3X$ (X = Sn and Ge) are antiferromagnetic at room temperature, with spins residing inside kagome planes of the crystal[18,43]. When the magnetic field is perpendicular to these planes, neither the magnetization nor the Hall resistivity show a jump. Previous studies have assumed that non-trivial topology of the electronic wave-function does not reveal itself in this configuration due to symmetry constrain of the planar spin texture[6,16–18].

In this work, by measuring the transverse electric and thermoelectric coefficients up to 14 T, we show that the in-plane field-linear Hall number of $Mn_3Sn$, is five times larger than what is expected from the carrier density and the Nernst signal is two orders of magnitude larger than what expected given the mobility and the Fermi energy of the system. We reveal an additional hidden component dominating the ordinary signals in both cases. Our theoretical calculations reveal that out-of-plane spin canting[44] induced by the magnetic field leads to nonzero spin chirality and simultaneously generates Berry curvature by gapping Weyl nodal lines unrecognized before. Our results present a unified mechanism between the real-space and momentum-space Berry phases as the origin of this unusual field-linear Hall effect.

## Results
### Field induced linear anomalous Hall effect

Figure 1 shows how drastically the Hall resistivity in $Mn_3Sn$ depends on the configuration. When the magnetic field, parallel to the kagome planes, is swept from negative to positive values (Fig. 1a), the $\rho_{zy}$ Hall resistivity displays a jump (Fig. 1c). However, when the field is perpendicular to the planes (Fig. 1b), the $\rho_{xy}$ Hall response is perfectly linear as a function of magnetic field. Figure 1c shows this drastic difference between out-of-plane ($\rho_{zy}$) and in-plane ($\rho_{xy}$) Hall resistivities at 200 K.

Scrutinizing $\rho_{zy}$, one can see that it consists of two parts. The first is the anomalous Hall resistivity $\rho_{zy}^A$, which has a spontaneous amplitude of 4.2 μΩcm at 0 T. The second is the ordinary Hall resistivity, $\rho_H^O$, which attains the amplitude of 0.77 μΩcm, when the field is swept from 0 T to 14 T. It corresponds to a carrier density of $n = 1.14 \cdot 10^{22}\,cm^{-3}$, consistent with previous reports[18,22] and the theoretically calculated carrier density[27]. At the first sight, $\rho_{xy}$, which is linear in magnetic field, looks like an ordinary Hall response. However, its amplitude, as large as 3.9 μΩcm at 14 T, is five times larger than $\rho_H^O$ and is incompatible with the large carrier density of the system. Note that invoking the presence of carriers of both signs would pull down the Hall number and does not provide a solution for the puzzle of an anomalously large Hall number.

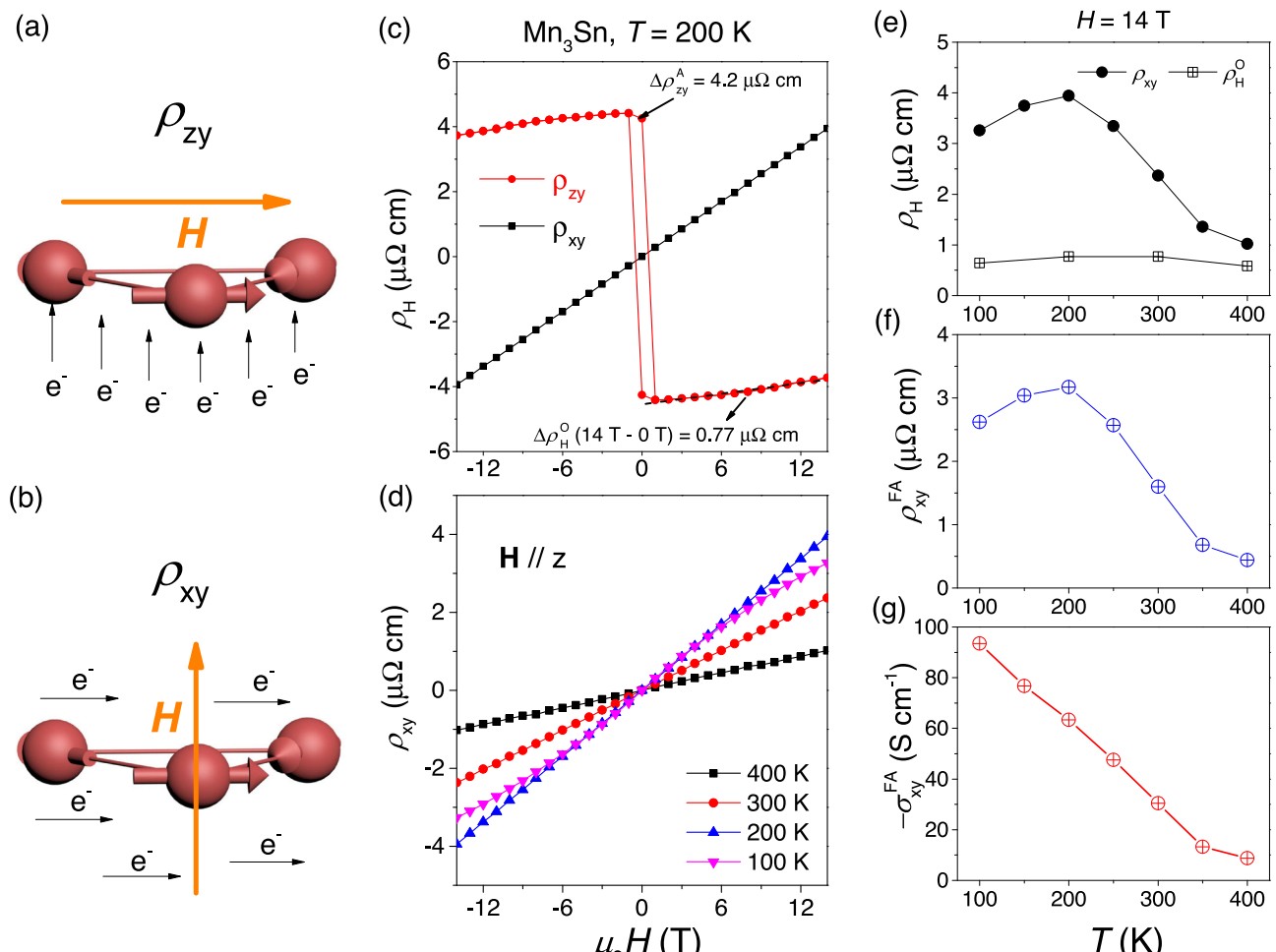

**Fig. 1 | Field induced anomalous Hall effect. a, b** Different Hall configurations. $\rho_{zy}$, the out-of-plane Hall configuration with the field along x axis. $\rho_{xy}$, the in-plane Hall configuration with the field along z axis. **c** Comparison of out-of-plane ($\rho_{zy}$) and in-plane ($\rho_{xy}$) Hall responses. $\rho_{zy}$ consists of two parts, the anomalous Hall resistivity $\rho_{zy}^A$ with a spontaneous value of 4.2 μΩcm at 0 T, and the ordinary Hall resistivity $\rho_H^O$ with a value of 0.77 μΩcm when the field is swept from 14 T to 0 T. $\rho_{xy}$, looking like ordinary Hall effect attains 3.9 μΩcm at 14 T, exceeding $\rho_H^O$ by far. **d** Field dependence of $\rho_{xy}$ with temperature varying from 100 to 400 K. **e** Temperature dependence of $\rho_{xy}$ and $\rho_H^O$ at 14 T. **f** Temperature dependence of field-induced linear anomalous Hall resistivity $\rho_{xy}^{FA}$. **g** Temperature dependence of field-induced linear anomalous Hall conductivity $\sigma_{xy}^{FA}$ (see the Supplementary Information for calculation details). It is monotonously increasing with cooling.

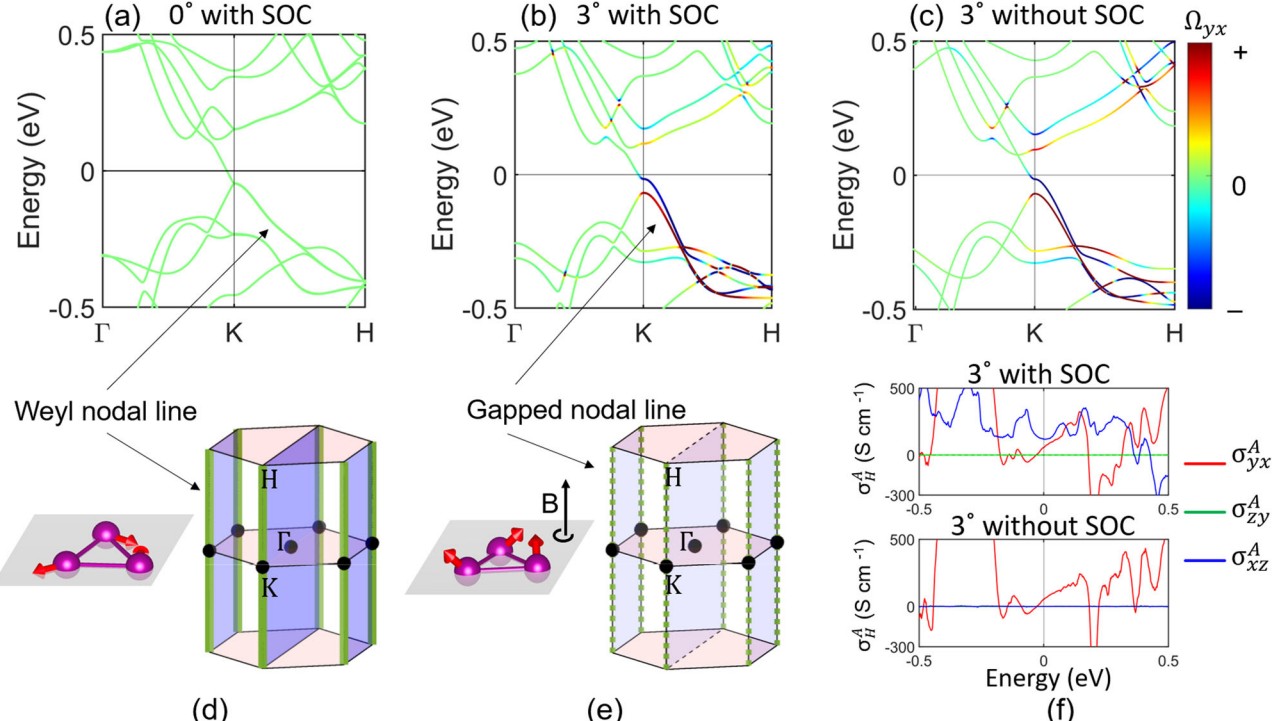

**Fig. 2 | Spin canting gaps out Weyl nodal lines and pushes Berry curvature to the Fermi surface. a, b** Band structures without and with spin canting (3°), respectively. **c** Band structure with 3° canting but excluding SOC. The color bar represents the amplitude of Berry curvature $\Omega_{yx}$. Without spin canting, there is a doubly-degenerate (weakly gapped by SOC) Weyl nodal line dispersing along the $K-H$ axis (including the $K$ point) in the Brillouin zone, as indicated by the solid green line in **d**. Spin canting significantly gaps out the nodal line, as indicated by the dashed green line in **e**, and induces giant Berry curvature $\Omega_{yx}$ on split bands in **b** and **c**. The mirror planes [$M_x$, blue planes in **d**] without spin canting forces $\Omega_{yx} = 0$ inside the plane. The Fermi energy is shifted to zero. **f** The Fermi energy-dependent anomalous Hall conductivities for 3° canting. $\sigma^A_{yx}$ is dominantly contributed by the spin chirality while $\sigma^A_{zy}$ relies on SOC.

We will see below that the large amplitude of $\rho_{xy}$ includes a hidden topological component.

Figure 1d shows the field dependence of $\rho_{xy}$ at different temperatures varying from 100 to 400 K. Note the drastic diminished amplitude of the 400 K curve. Figure 1e shows the temperature dependence of $\rho^O_H$(14 T) and $\rho_{xy}$(14 T). As expected, $\rho^O_H$(14 T) is flat over a wide temperature range, consistent with its identified origin. Indeed, the Fermi surface topology and the carrier density do not vary with cooling. On the other hand, $\rho_{xy}$(14 T) shows a strong temperature dependence and decreases rapidly with warming. As the Néel temperature ($T_N = 420K$) is approached, it becomes close to $\rho^O_H$. This implies that the temperature-independent ordinary Hall effect is almost isotropic and $\rho_{xy}$(14 T) acquires another component below $T_N$. This additional component, which we call field-induced linear anomalous Hall resistivity (FILAHE) $\rho^{FA}_{xy}$, be quantified by extracting $\rho^O_H$ from $\rho_{xy}$(14 T). Figure 1f shows this FILAHE, $\rho^{FA}_{xy}$. Its conductivity counterpart, $\sigma^{FA}_{xy}$, is shown in Fig. 1g. It is monotonously increasing with cooling.

Before presenting a theoretical explanation, let us briefly notice that the anomalous nature of the Hall response produced by on out-of-plane magnetic field was overlooked by previous studies. In 2016, Nayak et at.[19] measured the Hall responses of Mn$_3$Ge up to 5 T for three different configurations and found that when the field is along the z direction, the slope is significantly larger. More recently, a review of the transport properties of Mn$_3$X (X = Sn, Ge) contrasted the absence of hysteresis in $\rho_{xy}$ with its presence in $\rho_{yz}$ and $\rho_{zx}$[45].

## Spin canting gaps out Weyl nodal lines and pushes Berry curvature to the Fermi surface

Assuming that magnetic field modifies the spin structure can provide an explanation for this finite $\sigma^{FA}_{xy}$. Such an assumption is supported by a recent torque magnetometry study in Mn$_3$Sn[44]. Magnetic field favors alignment of spins along its orientation. When it rotates in the basal plane, this Zeeman effect would enter a competition with three other energy scales of the system (Heisenberg, Dzyaloshinskii-Moriya and Single-ion-anisotropy)[31] in order to generate a non-trivial twist of spins[46], giving rise to additional odd terms in the magnetic free energy[44]. Modification of the spin orientation by an *out-of-plane* magnetic field provides a hidden source of Hall response.

In symmetry analysis, the AHE XY component ($\sigma^A_{xy}$) without spin canting is strictly prohibited because the combined symmetry ($\mathcal{T}\mathcal{M}_z$) by time-reversal ($\mathcal{T}$) and mirror reflection ($\mathcal{M}_z : z \to -z$) constrains the Berry curvature $\Omega_{xy}$ to be opposite between $(k_x, k_y, k_z)$ and $(-k_x, -k_y, k_z)$. Further, $\sigma^A_{xy}$ vanishes also because a vertical glide mirror $\widetilde{\mathcal{M}_x} \equiv \{\mathcal{M}_x|\frac{c}{2}\}$ forces opposite $\Omega_{xy}$ between $(k_x, k_y, k_z)$ and $(k_x, -k_y, k_z)$. In the band structure, there is a seemingly crossing point at the $K$ point slightly below the Fermi energy (Fig. 2a), which was recognized as a Weyl point in some earlier studies. However, it is impossible to host a Weyl point at $K$, because the mirror plane would reverse the Weyl point chirality. At $K$, A tiny gap is actually opened by spin-orbit coupling (SOC). Without SOC, we find a doubly-degenerate nodal line, referred to as the Weyl nodal line, along the $K-H$ axis in the Brillouin zone (see Supplementary Fig. 2). Then, SOC induces tiny hybridization gaps along the nodal line. Along $\Gamma-K-H$, however, $\Omega_{xy}$ is always zero (Fig. 2a) because of the $\mathcal{M}_x$ mirror plane.

After spin canting or spin chirality appears and breaks both $\widetilde{\mathcal{M}_x}$ and $\mathcal{T}\mathcal{M}_z$ symmetries, the XY AHE emerges. In the band structure, the symmetry reduction significantly enlarges the energy gap along the Weyl nodal line. More importantly, spin canting generates a large Berry curvature $\Omega_{xy}$ along the nodal line. Because the gap is large, $\Omega_{xy}$ is smeared out and reaches the Fermi surface, as shown in Fig. 2b. Therefore, the net Berry curvature at the Fermi surface, and its associate anomalous Hall conductivity $\sigma^A_{xy}$, emerge. On the other hand,

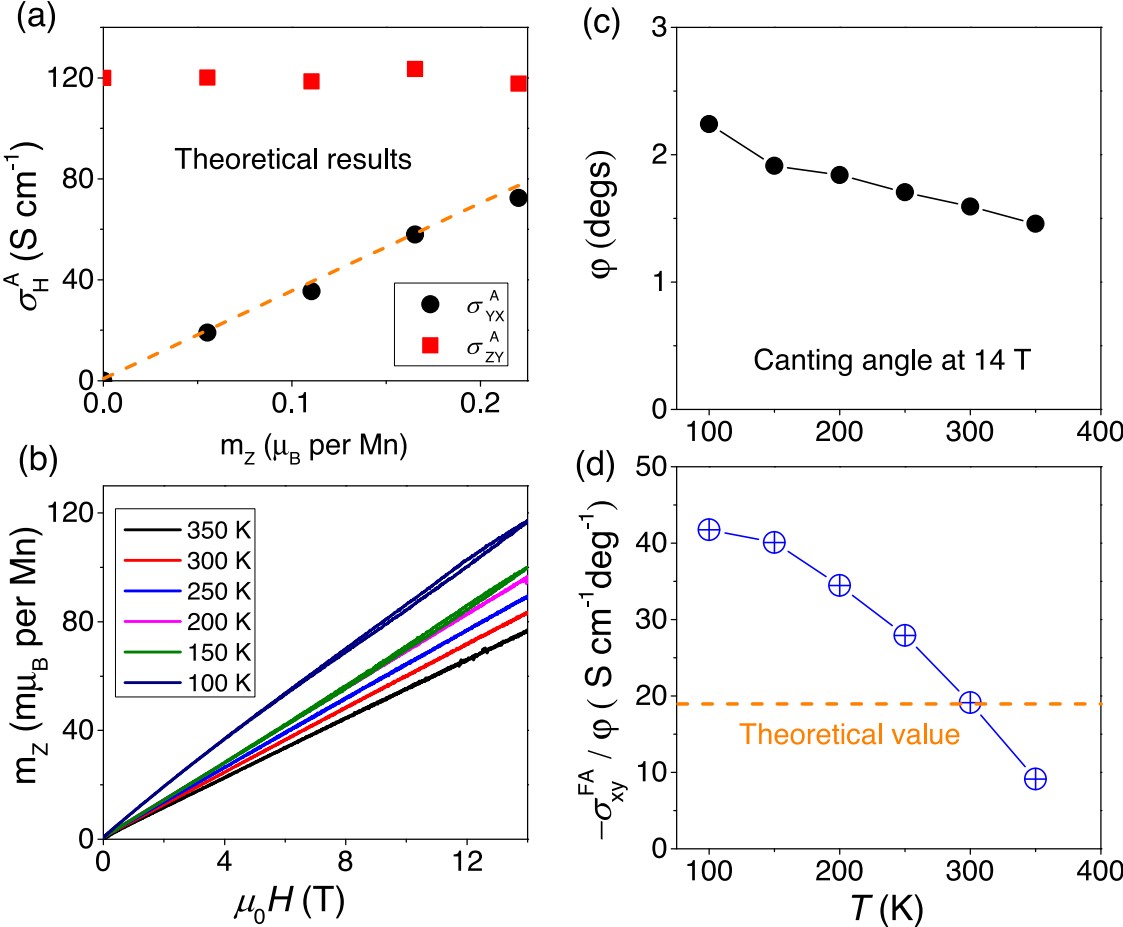

**Fig. 3 | Comparison between theory and experiment. a** Comparison of the theoretical calculated anomalous Hall conductivity in different Hall configurations (YX and ZY). The $\sigma_{zy}^A$ shows a flat behavior, but the $\sigma_{yx}^A$ shows a fast increasing with the $m_Z$. **b** The field dependent canting moment ($m_Z$) with temperature varying from 350 to 100 K. **c** Temperature dependence of the canting angle $\phi$, calculated from data in **b** and by formula arcsin($m_Z/3\mu_B$). **d** The ratio of $\sigma_{xy}^A$ to the canting angle at different temperatures.

the scalar spin chirality ($\chi$) is linearly proportional to canting spin angle, e.g., $\chi = S_1 \cdot (S_2 \times S_3)$ where $S_{1,2,3}$ are spins in a triangle, in the small tilting limit. Spin chirality can generate a real-space Berry phase when an electron hops between spin sites. To validate the role of spin chirality in AHE, we calculated the band structure and Berry curvature by excluding SOC. As shown in Fig. 2c, one can find that the Berry curvature and band structure remain almost the same as the SOC case (Fig. 2b), where SOC merely lifts some degeneracy in the band structure. In the absence of SOC, $\sigma_{yx}^A$ is nearly unchanged while $\sigma_{yz}^A$ becomes zero (see Fig. 2f). Therefore, the spin chirality-induced Hall response ($\sigma_{yx}^A$) coincides (rather than adds up to) with the AHE derived from the Berry curvature. In addition, Weyl points may exist near the nodal line gap due to accidental band crossing, for example, among bands near −0.3 eV which marginally affect the Fermi surface. The nodal line gap near the $K$ point is the main, direct Berry curvature origin to the AHE observed, which is further indicated by the Fermi energy-dependence of the anomalous Hall conductivity in Fig. 2f (also Supplementary Fig. 3). In addition, we showed the spin-canting induced anomalous Nernst coefficient in Supplementary Fig. 4.

**Comparison between theory and experiment**
The canting angle of spins cannot be directly probed by our experimented. However, a reasonable assumption is to compare the theoretical canting with the experimentally resolved field-linear magnetization when the field is oriented along the z-axis[31,44]. As seen in Fig. 3b, this magnetization changes from 80 to 120 m$\mu_B$ per Mn atom at

14 T between 100 to 400 K. Assuming that this is entirely caused by the field-induced canting of spins (i.e., neglecting any zero-field canting of spins), it would correspond to a canting angle ($\phi$) of the order of ~2 degrees. Figure 3c shows the canting angle estimated in this way at different temperatures. Figure 3d shows the evolution of the ratio $\sigma_{xy}^A/\phi$ at different temperatures. One can see that there is an agreement in order of magnitude. However, the experimentally resolved temperature dependence of $\sigma_{xy}^A/\phi$ is not captured by our model. This points to a missing ingredient, yet to be identified. We note that the canting angle of spins remains unmeasured. Our estimation was based on the amplitude of the out-of plane magnetization, which in presence of finite transverse magnetic susceptibility, may not be strictly accurate. Future studies of spin texture in presence of magnetic field may settle this discrepancy.

It is worth noting that the cluster multipole theory may also apply to our findings. Suzuki et al.[47] proposed that the in-plane octupole moment $T_x^y$ ($T_y^\gamma$) is lower than the three-dimensional octupole moment $T_{xyz}$ ($T_z^\beta$) in each cluster, and the neighboring clusters have ferromagnetic (net) and antiferromagnetic (vanished) alignments respectively. Out-of-plane spin canting may induce a finite $T_{xyz}$ ($T_z^\beta$) between neighboring clusters and generates FILAHE.

**Field induced linear anomalous Nernst effect**
Our interpretation is further supported by our measurements of the Nernst effect. Figure 4a, b shows the field dependence of the Nernst signal $S_{xy}$ and the transverse thermoelectric conductivity $\alpha_{xy}$ up to 14 T

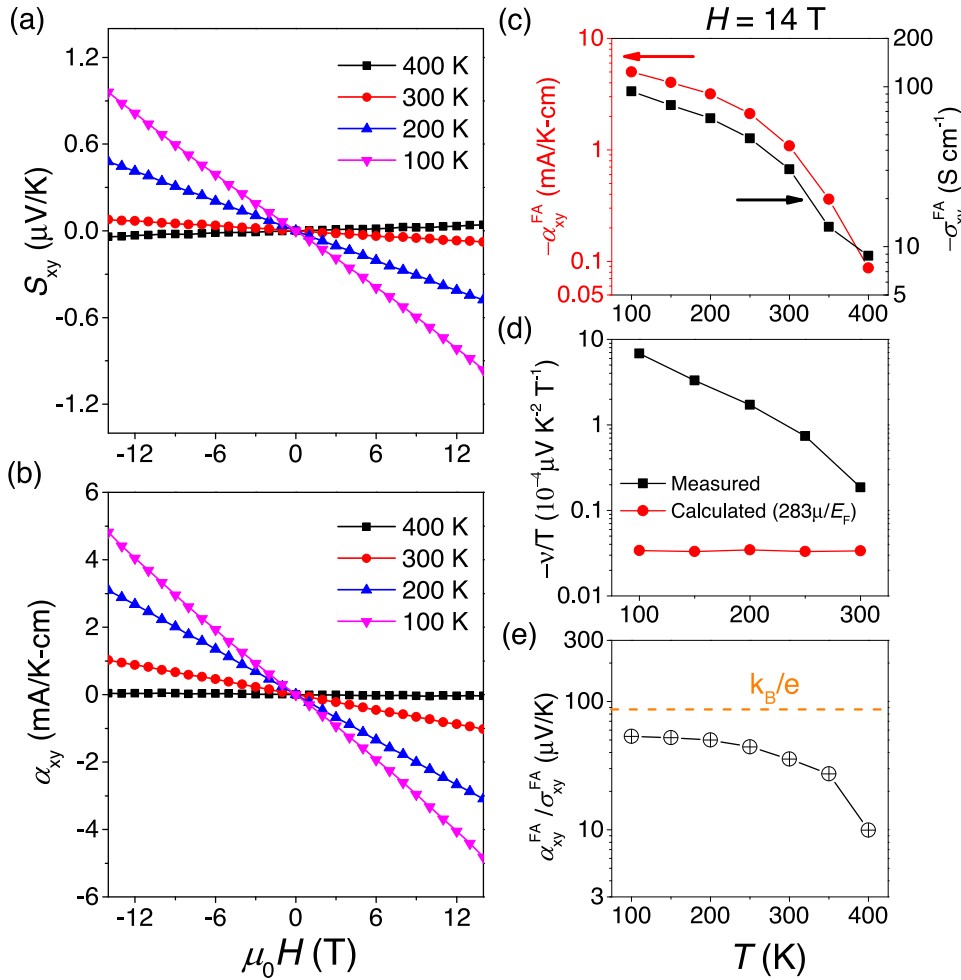

**Fig. 4 | Field induced anomalous Nernst effect. a, b** Field dependence of $S_{xy}$ and $\alpha_{xy}$ up to 14 T, with temperature varying from 100 to 400 K. **c** Comparison of temperature dependence of $\sigma_{xy}^{FA}$ and $\alpha_{xy}^{FA}$ at 14 T. **d** Temperature dependence of the Nernst coefficient ($\upsilon = S_{xy}/H$) divided by temperature. Measured Nernst signal (the black squares) with the largest value of $6.85 \times 10^{-4}$ $\mu VK^{-2}T^{-1}$, becomes two orders of magnitude larger than the estimated ordinary Nernst signal (the red circles) $3.45 \times 10^{-6}$ $\mu VK^{-2}T^{-1}$ estimated by $283\mu/E_F$, here $\mu$ and $E_F$ are carrier mobility and Fermi energy. **e** Temperature dependence of the ratio of off-diagonal field induced anomalous Hall and Nernst conductivity ($\alpha_{xy}^{FA}/\sigma_{xy}^{FA}$).

as the temperature changes from 100 to 400 K. As seen in Fig. 4c, the Nernst response, like the Hall response but more drastically, decreases with warming and approaching the Néel temperature. At 400 K, the former disappears, but the latter remains finite. As seen in Fig. 4d, the measured Nernst signal becomes two orders of magnitude larger than what is theoretically expected for the amplitude of the ordinary Nernst signal and experimentally observed in a variety of solids[48].

This, the amplitude of the Nernst response is incompatible with an ordinary origin. On the other hand, it does correspond to what is expected in a topological picture of the amplitude of anomalous transverse thermoelectric conductivity[49,50]. Indeed, in topological magnets, the amplitude of the anomalous Nernst signal anti-correlates with mobility[49] and the amplitudes of the anomalous Hall and Nernst conductivities correlate with each other[50]. In the case of the field-induced signals observed here, we found that $\alpha_{xy}^{FA}/\sigma_{xy}^{FA}$, varies from 10 $\mu$V/K at 400 K to 53 $\mu$V/K at 100 K, approaching $k_B/e = 86$ $\mu$V/K. Such a behavior has been observed in a variety of other magnets displaying anomalous transverse response[50].

In summary, we found a new variety of anomalous Hall effect and its Nernst counterpart induced by magnetic field. We showed that spin chirality induces Berry curvature on the Fermi surface by gapping the Weyl nodal line, resulting in the field-induced linear anomalous transverse response.

## Methods

### Samples

The centimeter-size $Mn_3Sn$ crystal was grown by the vertical Bridgman technique[27]. Firstly, the raw materials (99.999% Mn, 99.999% Sn) with the molar ratio of 3.3 : 1 were heated up to 1100 °C for the precursor crystal growth. Secondly, the precursor crystal power was put in an alumina crucible and sealed in a quartz tube and hung in a vertical Bridgman furnace for the single crystal growth. The growth procedure was repeated three times with different rates such as 2, 2 and 1 mm/h to purify the crystal. Using the energy dispersive X-ray spectroscopy (EDX), the stoichiometry of single crystal was found to be $Mn_{3.22}Sn$, close to but slightly below the ratio of the raw materials[37]. Finally, the large size single crystal was cut to desired dimension sample, such as 2.5 mm × 1.6 mm × 0.1 mm used in this work, by a wire saw.

### Measurements

All transport experiments were performed in a commercial measurement system (Quantum Design PPMS). Electric transport responses were measured by a standard four-probe method using a current source (Keithley6221) and a DC-nanovoltmeter (Keithley2182A). Thermoelectric transport responses were measured at a high vacuum environment, using a 4.7 kΩ chip resistor for the heater, a copper plate for the heat-sink, and a difference type E thermocouples for detecting

the temperature difference. Magnetization was measured by the vibrating sample magnetometer (VSM) mounted on PPMS. All measurements were performed on the same sample.

## Calculations

We performed the density-functional theory (DFT) calculation follows in the framework of the generalized gradient approximation with the Vienna ab intio package. We employed the PBE-D2 method to describe vdW interaction . Spin-orbit coupling (SOC) was included in all calculations. The magnetic ground state of the $Mn_3Sn$, which has an antichiral triangular inplane spin structure with $3\mu_B$ magnetic moment for each Mn atom as same as in the experiment (Fig. 2b). Start from the inplane spin structure as canting angle 0°, we tilted the spin-direction to out of plane direction uniformly to imitate applying the magnetic field in the experiment (Fig. 2d). After the magnetic state relaxation we can find local minimum spin state with finite $m_z$ magnetic moment.

We have projected the DFT Bloch wave function into Wannier functions to construct an effective Hamiltonian ($\hat{H}$) to evaluated the anomalous Hall conductivity:

$$\sigma_{ij}^k(\mu) = -\frac{e^2}{\hbar} \int_{BZ} \frac{d\mathbf{k}}{(2\pi)^3} \sum_{\epsilon_n < \mu} \Omega_{ij}^z(\mathbf{k}) \tag{1}$$

$$\Omega_{ij}^k(k) = i \sum_{m \neq n} \frac{\langle n|\hat{v}_i|m\rangle \langle n|\hat{v}_j|m\rangle - (j \leftrightarrow i)}{(\epsilon_n(\mathbf{k}) - \epsilon_m(\mathbf{k}))^2}. \tag{2}$$

$$\frac{\alpha_{xy}^A}{T}\Big|_{T \to 0} = -\frac{\pi^2 k_B^2}{3|e|} \frac{d\sigma_{xy}^A}{d\mu}, \tag{3}$$

Here $\mu$ is the chemical potential and $\epsilon_n$ is the eigenvalue of the $|n\rangle$ eigenstate, and $\hat{v}_i = \frac{dH}{\hbar dk_i}$ ($i = x, y, z$) is the velocity operator. A $k$-point of grid of $100 \times 100 \times 100$ is used for the numerical integration. Near the zero temperature, the anomalous Nearst coefficient follows the Mott relation.

## Data availability

The data that support the findings of this study are available from the corresponding author upon reasonable request.

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

## Acknowledgements

This work was supported by The National Key Research and Development Program of China (Grant No. 2022YFA1403503), the National Science Foundation of China (Grant No. 12004123, 51861135104 and 11574097) and the Fundamental Research Funds for the Central Universities (Grant no. 2019kfyXMBZ071). B.Y. acknowledges funding from the European Research Council (ERC) under the European Union's Horizon 2020 research and innovation programme (ERC Consolidator Grant "NonlinearTopo", No. 815869). K.B. was supported by the Agence Nationale de la Recherche (ANR-18-CE92-0020-01; ANR-19-CE30-0014-04). B.Y. and K.B. acknowledge a research grant from the Potter's Wheel Foundation/Weizmann-CNRS collaboration program. X.L. acknowledges the China National Postdoctoral Program for Innovative Talents (Grant No. BX20200143) and the China Postdoctoral Science Foundation (Grant No. 2020M682386).

## Author contributions

X.L., Z.Z., K.B. and B.Y. conceived of and designed the study. X.L. per-formed the transport measurements. J.K. and B.Y. performed the theo-retical calculations. X.L., J.K., Z.Z., K.B. and B.Y. analyzed the data. X.L., Z.Z., K.B. and B.Y. wrote the manuscript with assistance from all the authors.

## Competing interests

The authors declare no competing interests.
