## [Peer Review File · Nature Communications]

Reviewers' Comments:

Reviewer #1:

Remarks to the Author:

This work reports experimental and theoretical study of field-induced anomalous Hall effect in Kagome antiferromagnetic Mn₃Sn. Understanding the anomalous Hall effect's origin in antiferromagnetic is essential to designing new novel materials applicable to spintronics and thermoelectric devices. The results obtained in this manuscript support the previous theoretical prediction that nodal lines are pretty crucial to producing significant anomalous Hall effect and Nernst effect.

I think that the problems addressed and results are interesting, however, I would like to bring the attention of the Authors to the following points that make the present manuscript not suitable for publication in Nature Communications.

(1) The authors claimed that "The nodal line gap near the K point is the main, direct Berry curvature origin to the AHE observed". However, it is not supported by the present theoretical calculations. To support the claim, the authors should confirm it by computing the chemical potential dependence of anomalous Hall conductivity.

(2) The authors claimed that "Our work reveals intriguing unification of real-space Berry phase from spin chirality and momentum-space Berry curvature" However, it is not supported by the present theoretical calculations. To discuss the real-space Berry phase from spin chirality, the authors should provide computational results of anomalous Hall conductivity without spin-orbit coupling.

(3) The author should discuss the cluster multipole theory which explain anomalous Hall conductivity in Mn₃Sn, PRB 95, 094406 (2017).

(4) For the experimental aspect, the author should comment on the difference between the present study and Nayak et al. Sci. Adv. 2016;2:e150187. The authors also should comment on also other related review such as Chen et al. Nature Communications volume 12, 572 (2021).

Reviewer #2:

Remarks to the Author:

The paper "Identifying Berry curvature as the driver of field-linear in-plane Hall response of Mn₃Sn" focused on the Weyl antiferromagnet Mn₃Sn and measured the anomalous Hall transport in two ways (ρ_{zy} and ρ_{xy}) up to a relatively high magnetic field ~ 14 T and observed "linear" dependence of magnetic field in case of ρ_{xy} . Owing to the measurement at a high field, the authors could extract the ordinary component and identify the field-induced linear AHE (FILAHE). They also found that this FILAHE is significantly larger than naïve expectation assuming the ordinary Hall effect and they pointed out that it is coming from the additional large Berry curvature component arising from the spin canting from the coplanar 120-degree structure. They also measure the anomalous Nernst effect and observed the significant behaviors which have a common origin to AHE.

The authors pointed out the important aspect of transport properties in the Weyl antiferromagnet Mn₃Sn, with reliable experimental data and theoretical analysis. However, I have some confirmation and questions on the interpretation, which will strike the main conclusion of this paper. Therefore, I do not recommend its publication to nature communication at present. A and B is the important questions and suggestion. C is just a minor suggestion.

A. On the FILAHE observation

A-1. I understand that the origin of the linearity of FILAHE is due to the magnetization curve with small hysteresis. [e.g. Nakatsuji, Kiyohara, Higo, Nature 512, 213 (2015)] Is this correct?

A-2. I also want to confirm the reason why the previous experiments failed to extract FILAHE. I interpreted that the measurements and analysis with respect to the high field are important. Is this right?

A-3. The above high-field nature and comparison with previous studies should be discussed more clearly in the main text.

B. On the scalar spin chirality scenario

B-1. The authors discussed FILAHE as the effect of the scalar spin chirality. But I cannot agree with this since one may also explain the present Berry curvature by (simply) the effect of uniform magnetization (exchange splitting) and spin-orbit interaction since the (infinitesimal) uniform component of magnetization can open the Weyl gap and induce the nontrivial Berry curvature distribution. The authors also show the estimation of the spin canting (and I agree with this), but this is still consistent with the uniform magnetization scenario. Can the authors prove that the present Berry curvature is coming from the scalar spin chirality?

B-2. Another possibility is that the present FILAHE is coming from the combined effect of the scalar spin chirality and exchange splitting coupled with spin-orbit interaction. Can't authors consider this direction?

B-3. Suggestion: If the origin of FILAHE is a scalar spin chirality, we may expect a similar result by disregarding spin-orbit coupling [Only the exchange interaction between conduction electron and localized spin is necessary; See G. Tatara and H. Kawamura, *J. Phys. Soc. Jpn.* 71, 2613 (2002), and/or K. Ohgushi, S. Murakami and N. Nagaosa, *Phys. Rev. B* 62, R6065 (2000) for example]. Can the authors check this theoretically and/or experimentally?

C. Maybe the authors can theoretically show the energy dependence of the anomalous Hall conductivity (resistivity) and anomalous Nernst conductivity (coefficient). This will be beneficial for the readers.

Reply to reviewers

We thank both reviewers for their review and comments on our manuscript.

Reviewer #1 (Remarks to the Author):

This work reports experimental and theoretical study of field-induced anomalous Hall effect in Kagome antiferromagnetic Mn₃Sn. Understanding the anomalous Hall effect's origin in antiferromagnetic is essential to designing new novel materials applicable to spintronics and thermoelectric devices. The results obtained in this manuscript support the previous theoretical prediction that nodal lines are pretty crucial to producing significant anomalous Hall effect and Nernst effect. I think that the problems addressed and results are interesting, however, I would like to bring the attention of the Authors to the following points that make the present manuscript not suitable for publication in Nature Communications.

Reply:

We thank the referee for raising several important issues missing in our manuscript and deserving attention. However, let us recall the main message of our paper: We have discovered an Anomalous Hall Effect, perfectly linear in magnetic field and yet a consequence of Berry curvature. To the best of our knowledge, such an OBSERVATION has never been reported before, either in Mn₃Sn or in any other solid. The reviewer's interesting remarks put in to question a few details regarding our theoretical interpretation. However, we cannot detect any criticism putting into question the novelty of the observation of the possibility of an alternative interpretation.

(1) The authors claimed that "The nodal line gap near the K point is the main, direct Berry curvature origin to the AHE observed". However, it is not supported by the present theoretical calculations. To support the claim, the authors should confirm it by computing the chemical potential dependence of anomalous Hall conductivity.

Reply:

In the original Figure 2b, we showed the Berry curvature distribution of the Berry curvature. At the Fermi energy, there is substantial Berry curvature only near the K-point, presenting the direct theoretical evidence of the K-point gap contribution. For the nodal line gap dispersing along K-H, there is also significant Berry curvature compared to other bands. To make it more clear for the Reviewer, we show the energy dependence of AHC in Figure 2f, where one can find large AHC (σ_{yx}^A) in the nodal line energy range.

(2) The authors claimed that "Our work reveals intriguing unification of real-space Berry phase from spin chirality and momentum-space Berry curvature" However, it is not supported by the present theoretical calculations. To discuss the real-space Berry phase from spin chirality, the authors should provide computational results of anomalous Hall conductivity without spin-orbit coupling.

Reply:

We thank the Reviewer again for this expert advice. In the new Figure 2c, we show the band structure with Berry curvature distribution without including SOC, which agrees quantitatively with Figure 2b (the case with SOC). This is also reflected in the energy-dependence of AHC in Figure 2e. This indicates the role of spin chirality.

(3) The author should discuss the cluster multipole theory which explain anomalous Hall conductivity in Mn₃Sn, PRB 95, 094406 (2017).

Reply:

This is a valid point. Indeed, we failed to mention the Cluster Multiple Theory in the previous version. The new version includes this paragraph: “It is worth noting that the Cluster multipole theory may also apply to our findings. In 2017, Suzuki et al.^[47] proposed that the in-plane octupole moment T_x^γ (T_y^γ) is lower than the three-dimensional octupole moment T_{xyz} (T_z^β) in each cluster and the neighboring clusters have ferromagnetic (net) and antiferromagnetic (vanished) alignments. Out-of-plane spin canting may induce a finite T_{xyz} (T_z^β) between neighboring clusters and generates FILAHE.” We believe that including this possible interpretation will enrich our paper. It will not put in question the central finding, which is an anomalous Hall effect driven by field-induced spin canting.

(4) For the experimental aspect, the author should comment on the difference between the present study and Nayak et al. Sci. Adv. 2016;2:e150187. The authors also should comment on also other related review such as Chen et al. Nature Communications volume 12, 572 (2021).

Reply:

This also a valid point to consider. Both these papers do mention the Hall response caused by a magnetic field along the z-axis. However, neither of them notices that the Hall resistivity in this configuration cannot be explained by the semiclassical picture and is a case of anomalous Hall effect. In 2016, Nayak et al. measured the Hall resistivity of Mn₃Ge for three different configurations. But their discussion focused on in ρ_{yz} and ρ_{zx} , (and not ρ_{xy}). Similarly, in their review, Chen *et al.* wrote “The magnetic-field dependence of ρ_H is anisotropic; the sweep of the in-plane fields generates a substantial zero-field Hall component and a narrow hysteresis in ρ_{yz} and ρ_{zx} , whereas the out-of-plane field B||[0001] yields a linear-in-B response in ρ_{xy} without hysteresis. The in-plane-field responses, ρ_{yz} and ρ_{zx} , are unusually large for an AFM material and are comparable to the values found in strong ferromagnets.” One can see the contrast with our observation.

Following the reviewer’s comment, we added the following sentences in the new version:

“Before presenting a theoretical explanation, let us briefly notice that the anomalous nature of the Hall response produced by on out-of-plane magnetic field was overlooked by previous studies. In 2016, Nayak et al.^[18] measured the Hall responses of Mn₃Ge up to 5 T for three different configurations and found that when the field is along the z direction, the slope is significantly larger.”

“More recently, a review of the transport properties of Mn₃X (X = Sn, Ge) contrasted the absence of hysteresis in ρ_{xy} with its presence in ρ_{yz} and ρ_{zx} ^[45].”

Reviewer #2 (Remarks to the Author):

The paper “Identifying Berry curvature as the driver of field-linear in-plane Hall response of Mn₃Sn” focused on the Weyl antiferromagnet Mn₃Sn and measured the anomalous Hall transport in two ways (ρ_{zy} and ρ_{xy}) up to a relatively high magnetic field ~ 14T and observed “linear” dependence of magnetic field in case of ρ_{xy} . Owing to the measurement at a high field, the

authors could extract the ordinary component and identify the field-induced linear AHE (FILAHE). They also found that this FILAHE is significantly larger than naïve expectation assuming the ordinary Hall effect and they pointed out that it is coming from the additional large Berry curvature component arising from the spin canting from the coplanar 120-degree structure. They also measure the anomalous Nernst effect and observed the significant behaviors which have a common origin to AHE.

We thank the referee for this very accurate summary of our results.

The authors pointed out the important aspect of transport properties in the Weyl antiferromagnet Mn_3Sn , with reliable experimental data and theoretical analysis. However, I have some confirmation and questions on the interpretation, which will strike the main conclusion of this paper. Therefore, I do not recommend its publication to nature communication at present.

A and B is the important questions and suggestion. C is just a minor suggestion.

A. On the FILAHE observation

A-1. I understand that the origin of the linearity of FILAHE is due to the magnetization curve with small hysteresis. [e.g. Nakatsuji, Kiyohara, Higo, Nature 512, 213 (2015)] Is this correct?

Reply:

Fig.R1- Magnetization curves for three orientations of magnetic field. Both data sets show that when $B//z$, there is no hysteresis.

Before replying to the question A1, we invite the referee to have a look at Fig. R1. There is indeed a hysteresis in the magnetization of Mn_3Sn and it is related to the AHE, but the field-linear AHE we are talking about occurs when the magnetic field is along the z-axis for which there is no detectable hysteresis.

A-2. I also want to confirm the reason why the previous experiments failed to extract FILAHE. I interpreted that the measurements and analysis with respect to the high field are important. Is this right?

Reply:

As discussed in replying to the third question of reviewer 1, In fact, studies simply overlooked the fact that the amplitude of the Hall resistivity for $H//z$ does not correspond to the carrier density of this dense metal. Moreover, the field-dependence of the Hall resistivity for this orientation, which was perfectly linear in field, made it look like an ordinary Hall effect. Our field data is indeed important. It becomes clear that there is a large difference in linear slope between in-plane and out-of-plane orientations and therefore, the slope cannot be ascribed to the ordinary Hall response in

both directions. We think our conclusion that anomalous Hall effect can be linear in magnetic field will have a significant impact by changing the way experimentalists think about their Hall data.

A-3. The above high-field nature and comparison with previous studies should be discussed more clearly in the main text.

Reply:

As discussed in replying to third question of reviewer 1, we have revised the manuscript and inserted an account of the previous related studies and inserted two new references.

B. On the scalar spin chirality scenario

B-1. The authors discussed FILAHE as the effect of the scalar spin chirality. But I cannot agree with this since one may also explain the present Berry curvature by (simply) the effect of uniform magnetization (exchange splitting) and spin-orbit interaction since the (infinitesimal) uniform component of magnetization can open the Weyl gap and induce the nontrivial Berry curvature distribution. The authors also show the estimation of the spin canting (and I agree with this), but this is still consistent with the uniform magnetization scenario. Can the authors prove that the present Berry curvature is coming from the scalar spin chirality?

B-2. Another possibility is that the present FILAHE is coming from the combined effect of the scalar spin chirality and exchange splitting coupled with spin-orbit interaction. Can't authors consider this direction?

B-3. Suggestion: If the origin of FILAHE is a scalar spin chirality, we may expect a similar result by disregarding spin-orbit coupling [Only the exchange interaction between conduction electron and localized spin is necessary; See G. Tatara and H. Kawamura, J. Phys. Soc. Jpn. 71, 2613 (2002), and/or K. Ohgushi, S. Murakami and N. Nagaosa, Phys. Rev. B 62, R6065 (2000) for example]. Can the authors check this theoretically and/or experimentally?

Reply:

We thank the referee for the expert question and suggestion. Also see reply to the Reviewer 1, we show the band structure and AHE calculated without SOC in Figure 2c, which is almost the same as those including SOC in Figure 2b. It confirms that the dominant contribution is scalar spin chirality (mechanism B-3 mentioned by the Reviewer). In addition, the quantitative agreement of AHC at the Fermi energy (Figure 2f) indicates that SOC plays a marginal role.

C. Maybe the authors can theoretically show the energy dependence of the anomalous Hall conductivity (resistivity) and anomalous Nernst conductivity (coefficient). This will be beneficial for the readers.

Reply:

We show the energy-dependence of AHC in the new Figure 2f, and the ANC in the supplementary Figure S4.

Reviewers' Comments:

Reviewer #1:

None

Reviewer #2:

Remarks to the Author:

I found the authors' revisions of the manuscript are done satisfactorily for me.

The authors calculated the Hall conductivity in the absence of the spin-orbit coupling and confirmed the importance of the spin-chirality contribution of the FILAHE. Moreover, the authors clearly stated that the previous studies overlooked the FILAHE in the many studies in the Mn₃X series. They discuss the connection between the present results and the cluster multipole theory, which also makes the manuscript interesting. Now I think the manuscript becomes worth publishing to the nature communications.

Reply to reviewers

We appreciate that both reviewers devoted time to our manuscript.

Reviewer #2 (Remarks to the Author):

I found the authors' revisions of the manuscript are done satisfactorily for me.

The authors calculated the Hall conductivity in the absence of the spin-orbit coupling and confirmed the importance of the spin-chirality contribution of the FILAHE. Moreover, the authors clearly stated that the previous studies overlooked the FILAHE in the many studies in the Mn₃X series. They discuss the connection between the present results and the cluster multipole theory, which also makes the manuscript interesting. Now I think the manuscript becomes worth publishing to the nature communications.

We thank the reviewer for the recognition of our work and recommendation for publication.

Changes:

1. The section titles have been added.